# EGFP Reporters for Direct and Sensitive Detection of Mutagenic Bypass of DNA Lesions

**DOI:** 10.3390/biom10060902

**Published:** 2020-06-13

**Authors:** Marta Rodriguez-Alvarez, Daria Kim, Andriy Khobta

**Affiliations:** 1Unit “Responses to DNA Lesions", Institute of Toxicology, University Medical Center of the Johannes Gutenberg University Mainz, Obere Zahlbacher Str. 67, 55131 Mainz, Germany; mrodrigu@uni-mainz.de; 2Novosibirsk State University, 1 Pirogova St., 630090 Novosibirsk, Russia; kim.daria.nsk@gmail.com; 3Laboratory of Genome and Protein Engineering, SB RAS Institute of Chemical Biology and Fundamental Medicine, 8 Lavrentieva Ave., 630090 Novosibirsk, Russia

**Keywords:** DNA damage, DNA damage tolerance, damage bypass, translesion synthesis (TLS), transcriptional mutagenesis, mutation assay, reporter assay, host cell reactivation (HCR), enhanced green fluorescent protein (EGFP)

## Abstract

The sustainment of replication and transcription of damaged DNA is essential for cell survival under genotoxic stress; however, the damage tolerance of these key cellular functions comes at the expense of fidelity. Thus, translesion DNA synthesis (TLS) over damaged nucleotides is a major source of point mutations found in cancers; whereas erroneous bypass of damage by RNA polymerases may contribute to cancer and other diseases by driving accumulation of proteins with aberrant structure and function in a process termed “transcriptional mutagenesis” (TM). Here, we aimed at the generation of reporters suited for direct detection of miscoding capacities of defined types of DNA modifications during translesion DNA or RNA synthesis in human cells. We performed a systematic phenotypic screen of 25 non-synonymous base substitutions in a DNA sequence encoding a functionally important region of the enhanced green fluorescent protein (EGFP). This led to the identification of four loss-of-fluorescence mutants, in which any ulterior base substitution at the nucleotide affected by the primary mutation leads to the reversal to a functional EGFP. Finally, we incorporated highly mutagenic abasic DNA lesions at the positions of primary mutations and demonstrated a high sensitivity of detection of the mutagenic DNA TLS and TM in this system.

## 1. Introduction

Damage to DNA is usually repaired in cells within minutes; however, repair efficiency can be limited by a number of factors (for instance, peculiar structure of DNA modifications, their accessibility to DNA repair proteins, oversaturation of the cellular repair capacity by excessive damage loads, or impairment of the relevant repair pathway by a genetic, epigenetic or pharmacological mechanism). Therefore, damage tolerance is essential for cell survival, particularly under genotoxic stress conditions [1,2]. One of the most important DNA damage tolerance mechanisms is the ability of DNA polymerases to bypass DNA damage during replication [3]. As replicative DNA polymerases have very low capacities of DNA synthesis past the damaged bases, dedicated translesion synthesis (TLS) enzymes are required for copying damaged DNA [4,5]. However, since TLS capacity is incompatible with proofreading exonuclease activity, all DNA polymerases specialized in damage bypass have elevated error rates. This makes TLS an important source of DNA damage-induced mutagenesis and the major cause of point mutations found in cancers [6,7].

As replication machinery needs to deal with a variety of structurally different DNA lesions, higher organisms possess multiple TLS enzymes with characteristic DNA damage bypass efficiency and fidelity profiles, as exemplified by biochemical data available for human Y-family DNA polymerases [8,9,10,11,12]. However, physiological roles in the bypass of defined types of DNA modifications and the mechanism of selection of a particular TLS polymerase have been explored in far less detail. The analyses of mutations induced by specific types of DNA damage in the cellular context are extremely challenging or impossible in organisms with large genomes. Most of the relevant knowledge has come from artificially constructed TLS templates, which were delivered to suitable host cells for accomplishment of DNA synthesis over the damaged nucleotide. Thus, vectors carrying defined synthetic DNA modifications were designed either for integration into chromosomal DNA [3] or, more commonly, for allowing extrachromosomal TLS and repair followed by shuttling to bacteria and sequencing of mutations in the survived fraction of transformation-competent vector DNA [13,14,15,16]. The efficiency of detection of both faithful and mutagenic TLS can be immensely enhanced by accommodating the DNA lesion within a reporter gene [17,18]; however, for the fidelity assessment, TLS reporters described in the literature also require shuttling into bacteria [19]. Thus, a reporter system for the direct detection of mutagenic TLS in mammalian cells would bring extensive benefits to TLS research.

Another damage tolerance mechanism of emerging importance is the bypass of DNA lesions by transcribing RNA polymerase complexes [20,21]. Mounting evidence strongly suggests that persistent transcription blockage by DNA damage is implicated in degenerative disease [20]. Additionally, transcriptional bypass may also play a role as a causal factor of disease if accompanied with a high error rate. Studies demonstrated that unrepaired miscoding lesions lead to accumulation of mutant transcripts in the process called “transcriptional mutagenesis” (TM) [22,23]. When critical cancer genes were affected, TM caused functional alterations in the cellular growth and DNA damage response pathways, even in the absence of genetic mutations [22,24]. Consequently, it was proposed that transient phenotypic fluctuations in the crucial genome stability pathways may increase the chance of mutation by stimulating premature replication and, thus, promote carcinogenesis [25,26].

Plasmid constructs carrying defined DNA lesions at specific nucleotide positions were instrumental for revealing the phenotypic outcomes of TM in yeast and mammalian cells by activation of either the encoded indicator proteins [27,28,29] or the functional components of intrinsic cellular signaling pathways [22,24]. We therefore hoped that an actionable reporter approach would be equally useful for characterization of mutations introduced by DNA TLS. We recently described a new enhanced green fluorescent protein (EGFP) c.613C>T point mutation, which leads to the synthesis of a non-fluorescent EGFP Q205* truncated protein, and showed that any subsequent nucleotide substitution at the affected position results in reversion to a fluorescent phenotype, thus offering a sensitive reporter for detection of TM [30]. Importantly, the mutated base pair in the EGFP Q205* gene is flanked by tandem Bpu10I sites retained from the original EGFP coding sequence. This allows for a highly efficient targeted incorporation of synthetic DNA modifications into either transcribed (TS) or non-transcribed (NTS) DNA strands using commercially available pair of the strand-specific Bpu10I-derived nicking endonucleases [31,32]. In this work, we report application of the EGFP Q205* reporter to sensitive detection of mutagenic DNA synthesis over a single apurinic/apyrimidinic (AP) DNA lesion, incorporated opposite to a gap in the TS. Furthermore, based on screening of further 24-point mutations, we selected and validated a set of EGFP mutants suitable for direct analyses of DNA and RNA synthesis errors caused by damage affecting different DNA bases.

## 2. Materials and Methods

### 2.1. Reporter Constructs for Detection of TM

For the sake of coherence between the EGFP amino acid and nucleotide sequences, we used here the open reading frame-based rather than the protein-based numbering, with the initiator methionine numbered as 1, the A of the ATG codon numbered as c.1 and the complementary T as ts.1. This corresponds to the GenBank entries AAB08064 (protein) and U57609 (DNA). The pZAJ_5c vector encoding the functional EGFP protein [33] and the derived pEGFP_Q205* vector carrying the c.613C>T point mutation coding for the non-fluorescent truncated EGFP 1-204 protein, were described previously [30]. The procedure for site-specific incorporation of synthetic oligonucleotides into a single-stranded gap generated in the TS with the Nb.Bpu10I nicking endonuclease (Thermo Fisher Scientific Inc., St. Leon-Rot, Germany) was described previously [30]. Oligonucleotides 5′-TCAGGGCGGACT[THF]GGTGC-3′ and 5′-TCAGGGCGGACT[S-THF]GGTGC-3′ containing the synthetic AP lesion tetrahydrofuran with either phosphodiester (THF) or the nuclease-resistant phosphorothioate (S-THF) 5′-linkage were from BioSpring GmbH (Frankfurt am Main, Germany). The respective unmodified oligonucleotide 5′-TCAGGGCGGACTAGGTGC-3′ was from Eurofins Genomics (Ebersberg, Germany). The original pZAJ_5c vector accommodating the matched 5′-TCAGGGCGGACTGGGTGC-3′ synthetic oligonucleotide (Eurofins Genomics) was used as a reference for the original fluorescent EGFP. The quality of generated constructs carrying the specified modifications was controlled as described previously [30]. Representative results are shown in Appendix A. Incorporation of AP lesions into the newly generated pEGFP_A207P vector was performed by the same procedure, but synthetic oligonucleotides were 5′-TCAGGG[THF/S-THF]GGACTGGGTGC-3′ (with AP lesions) and 5′-TCAGGGGGGACTGGGTGC-3′ (unmodified control).

### 2.2. Reporter Constructs for Detection of Mutagenic TLS Templated by the AP Lesion

To construct a reporter for detection of mutagenic TLS, the Nt.Bpu10I nicking endonuclease (Thermo Fisher Scientific Inc., St. Leon-Rot, Germany) was used to generate the 18-nt gap in the pEGFP_Q205* vector (note the opposite strand specificity of the gapping reaction with respect to the procedure described for TM above). The procedure was essentially as described previously for pZAJ vector [32], except that the sequence of competitor oligonucleotide used to deplete the excised DNA fragment was 5′-GGGCGGACTAGGTGCTCA-3′. The resulting gap in the NTS was used to accommodate synthetic oligonucleotide 5′-TGAGCACC[THF]AGTCCGCCC-3′ containing the THF AP lesion (BioSpring GmbH) or the respective 5′-TGAGCACCTAGTCCGCCC-3′ unmodified oligonucleotide (Eurofins Genomics). Next, the opposite DNA strand was nicked at two sites with Nb.Bpu10I (2 U/microgram vector DNA) and the excised stretch of the TS depleted by incubation with excess of the complementary oligonucleotide 5′-GCACCTAGTCCGCCCTGA-3′ to generate 18-nt gap opposite to the strand carrying the lesion (Appendix A). Finally, gapped vector constructs were cleaned up using Amicon Ultra Ultracel 30 centrifugation devices (Millipore, Schwalbach am Taunus, Germany), as described previously [32]. The reference construct used to determine the gap repair efficiency in transfected cells was generated from the pZAJ_5c vector by the same procedure, but the synthetic oligonucleotide accomodated in the NTS was 5′-TGAGCACCCAGTCCGCCC-3′, which corresponds to the original nucleotide sequence of fully functional EGFP. Sequences of accessory oligonucleotides used for the strand depletion steps were the same as described previously [32].

For generation of the TLS reporter construct from the pEGFP_A207P vector, the procedure was the same as for pEGFP_Q205*, but in this case synthetic oligonucleotides were: 5′-GGGGGGACTGGGTGCTCA-3′ (depletion of the excised NTS fragment); 5′-TGAGCACCCAGTCC[THF]CCC-3′ (TLS template with AP lesion); 5′-TGAGCACCCAGTCCCCCC-3′ (unmodified control) and 5′-GCACCCAGTCCCCCCTGA-3′ (final depletion of the TS fragment).

### 2.3. Phenotypic Screening of the EGFP Mutants

The procedure for efficient integration of synthetic oligonucleotides with mismatched bases into circular DNA was described previously [34]. We started from the pZAJ_5c vector [33], which harbors the EGFP coding sequence from the pEGFP-C3 expression vector (GenBank accession number U57609) originally purchased from Clontech (Saint-Germain-en-Laye, France). Tandem nicks were generated in the TS with Nb.Bpu10I nicking endonuclease and the excised fragment was eliminated by incubation with a complementary competitor strand, as described previously [32]. The resulting 18-nt gap in the TS was filled by seamless ligation of 26 different synthetic oligonucleotides (all purchased from Eurofins Genomics): the matching 5′-TCAGGGCGGACTGGGTGC-3′ 18-mer (which reconstituted the original EGFP sequence) and 18-mers containing 25 non-synonymous single nucleotide substitutions at the underlined positions (which created a set of hybrid constructs with original EGFP sequence in the NTS and mutated TS). Integrity of the generated constructs and correct incorporation of the synthetic strands were verified by inhibition of ligation in the aliquots incubated in the absence of polynucleotide kinase, as described previously [32]. The respective controls are shown in Appendix A. The resulting constructs were transfected to DLD1 cells to analyze the EGFP fluorescence intensity.

### 2.4. Phenotypic Validation of Newly Identified EGFP Mutants

Vectors pEGFP_Q205P (containing the c.614A>C mutation), pEGFP_S206A (c.616T>G), pEGFP_S206Y (c.617C>A), and pEGFP_A207P (c.619G>C) were generated from the pZAJ_5c vector encoding the functional EGFP protein [33] by introducing the respective point mutations. Fluorescence intensities of mutant EGFP variants were measured in transfected HeLa cells, relative to the intensity of the original EGFP (measured in cells transfected with pZAJ_5c vector in parallel). As a non-fluorescent reference, cells were transfected with the pEGFP_Q205* vector encoding the truncated EGFP 1-204 variant described previously [30]. To assess phenotypes of secondary mutations affecting the selected nucleotides in the non-fluorescent mutants, the vectors pEGFP_Q205*, pEGFP_Q205P, pEGFP_S206Y, and pEGFP_A207P were nicked at the conserved tandem Nb.Bpu10I sites and the excised fragments of the TS were substituted in each mutant with synthetic oligonucleotides containing four different bases (N = G/A/C/T) at the primary mutation site. The procedure for strand substitution was the same as described under 2.3. The sequences of the oligonucleotides used for depletion of the excised TS fragment were 5′-GCACCTAGTCCGCCCTGA-3′ (in the case of pEGFP_Q205*); 5′-GCACCCCGTCCGCCCTGA-3′ (pEGFP_Q205P); 5′-GCACCCAGTACGCCCTGA-3′ (pEGFP_S206Y); and 5′-GCACCCAGTCCCCCCTGA-3′ (pEGFP_A207P). The sequences of the inserted oligonucleotides were 5′-TCAGGGCGGACTNGGTGC-3′ (in the case of pEGFP_Q205*); 5′-TCAGGGCGGACNGGGTGC-3′ (pEGFP_Q205P); 5′-TCAGGGCGNACTGGGTGC-3′ (pEGFP_S206Y); and 5′-TCAGGGNGGACTGGGTGC-3′ (pEGFP_A207P). Phenotypes of the secondary mutations were determined by transfection of the constructs into DLD1 cells.

### 2.5. Cell Culture, Transfection, and EGFP Expression Analysis

HeLa (human cervical carcinoma) and DLD1 (human colorectal carcinoma) cell lines were used as transfection hosts for EGFP expression analyses. Cells were propagated in DMEM high glucose medium supplemented with 10% fetal bovine serum. Twenty-four hours before transfection cells were seeded on 6-well plates (Nunc, Wiesbaden, Germany) at 300,000 HeLa cells/well and 400,000 DLD1 cells/well. Exponentially growing cells were transfected with the help of Effectene (Qiagen, Hilden, Germany). Equal amounts (400 ng) of the specified EGFP expression constructs and the pDsRed-Monomer-N1 vector (Clontech, Saint-Germain-en-Laye, France) were combined for transfections. Cells were fixed using 1% formaldehyde 24 h post-transfection and analyzed by flow cytometry, as described previously [35], using FACSCalibur^TM^ and the CellQuest^TM^ Pro software (Beckton Dickinson GmbH, Heidelberg, Germany). Untransfected cells were cut off by selective gating based on the DsRed expression prior to generation of the EGFP fluorescence (FL1-H) distribution plots and average EGFP signal per cell determined as the median of the distribution. For quantification of relative brightness of the EGFP mutants and for measurement of TM, the signal was normalized relative to the median fluorescence of the reference construct, coding for a fully functional EGFP, recorded in the same experiment. For evaluation of the TLS mutation rates, a template encoding the conventional EGFP was used to calculate the repair efficiency of the gap in the TS expressed as a ratio between the number of EGFP-positive cells and the total number of transfected cells. In parallel, the mutation rates were determined by the same approach in the mutant EGFP constructs containing the gap and either unmodified nucleotides or THF at the sites of primary mutations in the opposite DNA strand.

## 3. Results

### 3.1. Detection of Mutagenic Bypass of Abasic Site During RNA and DNA Synthesis Using EGFP Q205* Reporter

To generate a reporter for the direct detection of transcriptional mutagenesis (TM) and mutagenic DNA translesion synthesis (TLS) in human cells, we adopted the c.613C>T loss-of-function EGFP mutant encoding the truncated EGFP Q205* protein. Its DNA sequence allows for the strand-specific incorporation of oligonucleotides containing synthetic DNA lesions at defined positions (as explained in Materials and Methods and Appendix A) and, especially important, any single nucleotide substitution at the position 613 of the coding sequence leads to an amino acid which reconstitutes the EGFP fluorescence [30]. We used these features to design a transfection-based assay for direct detection of the TM and TLS occurring in human cells (Figure 1a).

For the detection of TM, a potentially miscoding DNA lesion needs to be inserted on the place of dA in the transcribed DNA strand (TS). Thereby, any misincorporated nucleotide in the messenger RNA opposite to the lesion would result in reversion to a fluorescent EGFP phenotype (Figure 1a, left arm). To ensure that the phenotypic change is driven by a nucleotide change in RNA, and not by mutation of DNA sequence acquired during propagation of vector DNA in the host cells, we used a vector that does not replicate in HeLa cells [31]. We reasoned that the same vector would be adaptable also for detection of TLS errors, but in this case the lesion has to be accommodated on the place of dT in the non-transcribed DNA strand (NTS) and a fragment of the opposite DNA strand needs to be removed to generate a single-stranded gap. Thereby, production of full-length EGFP transcripts in the host cells is only possible if preceded by re-synthesis of the missing stretch of the TS, which requires the lesion bypass (Figure 1a, right arm). After accomplishment of the gap repair, only expression constructs that had gained (by mutagenic TLS) a nucleotide substitution at the lesion site would yield a fluorescent EGFP protein.

Since abasic sites are extremely common and a highly mutagenic type of DNA damage, we used the AP lesion as a model DNA lesion for investigation of the bypass fidelity in HeLa cells. To detect TM, we analyzed EGFP fluorescence in cells transfected with constructs accommodating two types of synthetic AP lesions at the nucleotide 613 of the EGFP Q205* gene (Figure 1b). We used a synthetic tetrahydrofuran (THF) AP lesion and also its counterpart flanked by a 5′ phosphorothioate bond (S-THF), which confers resistance to the key base excision repair (BER) enzyme AP endonuclease 1 (APE1), as reported previously [30,36]. As expected, EGFP signal in cells transfected with the THF construct was negligible, because this type of AP lesion undergoes a very efficient BER resulting in reconstitution of the A:T base pair. In contrast, BER-resistant AP lesion (S-THF) showed a large proportion of reversion to a fluorescent EGFP phenotype in comparison to dA. This result indicates that transcriptional bypass of S-THF takes place with high rate of ribonucleotide misincorporation, which erases the premature termination codon and thereby restores the protein fluorescence. Thus, transcriptional mutagenesis over an abasic site can be readily detected in fully repair-proficient cells using the EGFP Q205* reporter.

To detect mutagenic DNA TLS, we substituted the nucleotide c.613T in the EGFP Q205* coding sequence for a synthetic abasic site (THF) and generated an 18-nucleotide gap in the transcribed DNA strand opposite to the lesion, as described under Materials and Methods (Section 2.2). For the TLS analyses, it was not necessary to use the phosphorothioate-protected (S-THF) lesion, because human APE1 does not efficiently cleave AP sites in single-stranded DNA (Appendix A). To monitor repair efficiency of the gap in the absence of damage in the single-stranded stretch of DNA, we generated and analyzed in parallel a wild-type EGFP construct containing an identical 18-nt gap in the TS. As judged by its expression, the gap repair efficiencies were heterogeneous across the cell population, with EGFP signal detected in approximately 83% of transfected cells (Figure 1c). We assumed that the repair efficiency of the gap must be the same in the EGFP Q205* construct encoding the non-fluorescent truncated protein. Nonetheless, the fraction of EGFP-positive cells in this case remained almost negligibly small (0.5 ± 0.3%), in accordance with the expectation of a very low rate of base substitutions spontaneously generated at the nucleotide 613 during re-synthesis of the missing DNA strand. Of note, the fraction of cells showing reversal to a fluorescent phenotype was strongly (by >30-fold) increased in the case of the c.613THF construct, as compared to c.613T, thus indicating that TLS over the THF lesion occurs with a high rate of nucleotide misincorporation. With these reporter constructs, we further were able to detect mutagenic TLS in several tumor as well as non-cancerous cell lines (data not shown and Appendix A). Intriguingly, albeit the reversion to fluorescent EGFP was universally observed, the mutant frequencies showed substantial variability between non-isogenic cell lines, as demonstrated by a representative data (Appendix A). This is likely to reflect not only differences in the TLS mechanism, but rather a variability of a broad range of factors, including DNA repair, recombination and the damage tolerance mechanisms.

In summary, the results shown in Figure 1 demonstrate usefulness of the EGFP Q205* reporter vector to efficiently detect both transcriptional mutagenesis and mutagenic translesion DNA synthesis in human cells.

### 3.2. Screening the EGFP Sequence for Candidate Single Nucleotide Substitutions Leading to the Loss of Fluorescence

Even though the EGFP Q205* reporter allows for a very sensitive detection of lesion-specific mutagenicity, its usefulness is limited to modifications affecting thymine (for detection of TLS) and adenine (for detection of TM). Moving towards a reporter system of a broader applicability, we sought other inactivating EGFP point mutations that would yield different types of base pairs in different NTS:TS orientations. Of note, the nucleotide sequence available for incorporation of synthetic DNA lesions, as confined by the available Bpu10I nicking sites, encodes a stretch of amino acids within the beta-strand 10. The target region lies in a close proximity to fluorophore in the folded EGFP, with at least two residues (T204 and S206) directly involved in functional molecular interaction with the fluorophore [37]. We therefore hoped that the likelihood of finding deleterious mutations by random mutagenesis of this region would be high. We compiled all single nucleotide substitutions possible within the sequence fragment flanked by the Bpu10I sites (Table 1). Of the 33 potential single nucleotide substitutions within the available 11-nucleotide window (four codons), seven did not confer an amino acid change and one was the c.613C>T nonsense transition mutation identified previously [30]. To rapidly examine whether any of the remaining 25 non-silent mutations would eliminate the EGFP fluorescence, we employed a simplified approach for the phenotypic screening of the candidate mutants. We took advantage of the Nb.Bpu10I nicking endonuclease to excise the TS fragment comprising the nucleotides 608–625 of the EGFP coding sequence in a vector encoding a functional EGFP protein and substituted it for synthetic DNA strands covering the whole spectrum of potentially useful mutations identified in Table 1, as described under Material and Methods (Section 2.3). In this way, we generated the set of hybrid EGFP constructs containing the innate NTS and various substituted nucleotides in the TS (Figure 2a). To prevent correction of the TS sequence by the DNA repair system in cells, we chose a mismatch repair-deficient DLD1 cell line as a suitable transfection host for the EGFP expression analyses.

Besides the 613C:A construct (which was expected to lose the fluorescence), seven other mismatches out of the 25 tested displayed more than a five-fold reduction of the EGFP fluorescence levels and thereby were considered as potential candidates for the loss-of-function mutations (Figure 2b). The repertoire of identified base substitutions with a loss-of-function potential covered the whole spectrum of nucleotides at the positions affected by the mutations—four with C:G (NTS:TS), one more with T:A (in addition to c.613C>T mutant, which was already available), and one each with G:C and A:T base pairs. One of the candidates (ts.610A>T) unintentionally dropped out of the screen. However, further phenotypic characterization of this nucleotide substitution turned out to be no longer necessary for the following reasons: firstly, the equivalent T204S amino acid substitution (conferred by the 611C:C mismatch) showed a residual EGFP signal, and secondly, the target T:A (NTS:TS) nucleotide pair was already covered by the available c.613C>T mutant.

For subsequent validation of the identified loss-of-fluorescence mutants, we prioritized those in which subsequent random nucleotide substitutions at the position affected by the primary mutation were more likely to restore the fluorescent phenotype. This could be predicted from reasonably high signals displayed by the respective neighbors in the preliminary screen (Figure 2b). We grouped mutations based on the resulting base pairs and identified the potential top candidates within each of the four groups (highlighted in bold in Table 1).

### 3.3. Validation of EGFP Mutants Carrying Inactivating Single Nucleotide Substitutions

We cloned four selected EGFP mutants (Q205P, S206Y, S206A and A207P) to validate their phenotypes by comparing brightness of the resulting proteins with conventional EGFP (encoded within the original pZAJ_5c vector) and with the Q205* variant identified previously as non-fluorescent. Three of the four new mutants (Q205P, S206Y and A207P) displayed no EGFP fluorescence in transfected HeLa cells. The S206A variant showed residual fluorescence, albeit of diminished brightness (relative expression = 0.331) (Figure 3a). We thus reasoned that the S206A mutant would be of limited usefulness for the detection of mutagenic TM or TLS; however, all other mutants were considered potentially suitable for this purpose.

For the efficient detection of mutagenesis at a target nucleotide in the TM or TLS assays, it would be essential to verify that subsequent base substitutions at the affected position efficiently restore the EGFP fluorescence. This was expected based on the results of the preliminary screening (Figure 2); however, the former approach was potentially prone to false positives, because construction of mismatched templates for the screen then departed from the fluorescent EGFP version. To test all 16 sequence variants more rigorously, we used a single-strand substitution approach applied previously, but inverted the detection principle by starting from inherently non-fluorescent EGFP. In each of the identified EGFP loss-of-fluorescence mutants, we replaced the TS fragment comprising nucleotides 608–625 for different synthetic strands, containing each of the four nucleotides at the positions of the primary base substitutions in the respective mutants (as described in the Methods Section 2.4.). An analysis of the EGFP fluorescence in DLD1 cells showed that all nucleotide substitutions in the TS at the primary mutation sites led to the at least partial restoration of the EGFP fluorescence (Figure 3b), whereas strand replacements not leading to nucleotide substitutions (NTS:TS 613T:A in Q205*; 614C:G in Q205P; 617A:T in S206Y; and 619C:G in A207P) all retained the loss-of-fluorescence phenotype. Based on the almost complete regain of EGFP fluorescence in the Q205*, Q205P and A207P mutants (Figure 3b and Table 2), we unequivocally predict that any base substitution mutation affecting the specified base pairs would be detectable with very high sensitivities in these mutants. In the case of the S206Y mutant, two of the secondary base substitutions (ts.617T>C and especially ts.617T>A) resulted in only partial recovery of the fluorescent signal, which would decrease detection sensitivity of these mutations proportionally. Still, even in the cases of base substitutions yielding the lowest levels of EGFP fluorescence, the signals clearly exceeded the EGFP S206Y background (by the factors of >40 and >6, for ts.617T>C and ts.617T>A, respectively). With these minor limitations, we conclude that all four newly generated EGFP mutants would be suitable as reporters for the direct detection of erroneous bypass of DNA lesions during transcription and DNA synthesis. The range of base pairs covered by the available mutants is applicable to analyses of mutagenic TLS at DNA damage affecting A, C or T in the coding DNA strand (Table 2). Conversely, TM can be analyzed for DNA lesions affecting A, G or T in the transcribed strand.

### 3.4. Detection of TM Using the A207P (c.619G>C) EGFP Mutant

To prove that TM can be sensitively detected at a nucleotide conferring a missense EGFP mutation, we further assessed the outcome of the highly miscoding S-THF AP lesion in the newly obtained EGFP A207P (c.619G>C) mutant. We incorporated synthetic THF and S-THF AP lesions at the nucleotide 619 in the TS (Figure 4a,b) and measured the resulting EGFP signal in transfected HeLa cells. As in the case of the Q205* (c.613C>T) mutant analyzed previously (Figure 1), unrepairable S-THF lesion caused very strong gain of the EGFP fluorescence as a result of TM (Figure 4c). The results thereby confirm extremely high miscoding potential of AP lesions, as previously observed in the Q205* (c.613C>T) reporter (Figure 1b). In both reporter systems, we observed recovery of the fluorescence intensity from zero to the level of 33–35% of the signal produced by fully functional EGFP protein, thus indicating that at least one third of all transcripts contained nucleotide substitutions at the lesion site. Since AP lesions undergo nucleotide excision repair (NER) in human cells, TM rates are even higher in NER deficient cells ([30] and unpublished results). As a demonstration of the extraordinary efficiency of repair of AP lesions in cells, it is important to note that the TM signal was fully eliminated when cells were transfected with the construct containing the BER-sensitive THF lesion. This is documented by the direct comparison with the control construct containing dG at the analyzed position (Figure 4c). In summary, the new reporter does not only allow for the extremely sensitive detection of TM, but it also offers a tool for the assessment of the repair capacity of cells with a very broad (at least 30-fold) dynamic range of detection.

### 3.5. Detection of Mutagenic TLS Using the A207P (c.619G>C) EGFP Mutant

The AP site is generally considered a “non-instructive” lesion and specificity of the deoxyribonucleotide incorporation during DNA synthesis over AP lesions in mammalian cells is a matter of controversy [9,14,38,39]. Some results indicated that different nucleotides are incorporated opposite to AP lesions in the template strand in a rather random fashion [38], whereas most reports suggested preferential (up to 85%) incorporation of dA, similar to the “A-rule” previously established for the SOS-induced replication in bacteria [40]. In the present study, we observed a pronounced regain of the EGFP signal by TLS of AP lesion in the EGFP Q205* reporter, as compared to the template containing thymine in the probed position (Figure 1c). This clearly indicated that dA is not the only nucleotide incorporated opposite to the AP lesion during repair synthesis over the 18-nucleotide gap, because adenine at the position 613 in the TS would restore the UAG translation termination codon. Therefore, to test whether the nucleotide incorporation opposite to AP lesions is random in human cells, it was instructive to analyze the mutation rate in a different reporter. We used the EGFP A207P mutant to generate gapped constructs containing either dC or THF at the nucleotide position 619 and analyzed the rates of reversion to the fluorescent EGFP in transfected HeLa cells, using a procedure analogous to the one used formerly for the EGFP Q205* reporter. As expected, the repair of the gap only occasionally led to reactivation of the EGFP fluorescence in the absence of the lesion (0.11 ± 0.02% cells). However, the fraction of EGFP-positive cells rose to 57.3 ± 2.8% (>500-fold) when dC in the NTS was substituted by the THF lesion (Figure 5). Thereby, the EGFP reactivation rate by AP site in the reporter that detects incorporation of any nucleotide except dG is substantially higher than in the EGFP Q205*, which is reactivated by incorporation of any nucleotide but dA (16.1 ± 4.4%, with an approximately 30-fold increase over the dT template, as shown in Figure 1). The result thus indicates that insertion of nucleotides opposite to AP lesion during DNA repair synthesis takes place in a non-random fashion and that incorporation of dA is strongly favored over dG human cells. Importantly, in the absence of the lesion, the repair rates of the gap were remarkably similar between these experiments (83.3 ± 0.9% and 77.8 ± 6.6%, as calculated based on the construct coding for the fully functional EGFP transfected in parallel). Thus, different EGFP reactivation rates between the A207P and Q205* mutants should be entirely attributed to differential incorporation of dA and dG at the damage site.

## 4. Discussion

The erroneous bypass of damaged nucleotides during DNA replication is a major source of mutations [6,7]. Some of the mutation signatures found in cancers have been attributed to environmental mutagens. However, as multi-layered resistance and repair mechanisms within the cell (comprising metabolism, DNA repair and DNA damage response) efficiently protect against experimental mutagenesis, many of the annotated mutation types are difficult to trace back to exposure to a particular carcinogen [41,42,43]. Along with the whole-genome sequencing approaches, the reporter-based strategies offer promising potential in addressing this fundamental problem [44]. An advantage of exogenously delivered vectors carrying specific types of DNA lesions is their suitability to investigate the outcomes of damage in DNA while avoiding cell exposure to deleterious doses of the damaging agent. This is especially valuable for the lesions that cannot be induced in the chromosomal DNA with high specificity and activate cytotoxic responses if they are present in the genome in significant amounts [45]. Methods have been developed for efficient incorporation of defined DNA lesions into vectors, which can be propagated in cells extrachromosomally [14,38,46,47,48,49]; and strategies for integration of DNA modified at precise sites into genomes are emerging [3,50,51]. In conjunction with ever growing availability of synthetic DNA modifications, replication- and transcription-competent extrachromosomal vectors, suitable for incorporation of structurally defined DNA modifications at defined positions, thereby open possibilities for investigation of miscoding properties of virtually any type of DNA lesion.

A straightforward and precise way for the targeted incorporation of synthetic DNA lesions into plasmid-based vectors is ligation of short synthetic DNA strands into gapped circular DNA, which can be conveniently generated using sequence-specific nicking endonucleases [31,52,53,54,55,56]. The use of nicking endonucleases for introducing modifications can be problematic in the protein coding regions of reporter genes, because conserved amino acid sequence restrains the possibilities for accommodation of the required recognition sites. Excitingly, however, we previously found out that the conventional EGFP coding sequence contained pre-existing tandem sites for the pair of nicking endonucleases Nb.Bpu10I and Nt.Bpu10I [32]. We realized that the opposite strand specificities of the available nickases provide an immediate advantage by allowing deliberate modification of either DNA strand. Consequently, the same reversible mutant containing an inactivating base substitution between the Bpu10I sites can be used as a reporter for either erroneous DNA TLS or TM simply by switching the nicking enzymes targeting either the NTS or TS, respectively. Moreover, the availability of nicking enzymes with opposite strand specificities allows precise and efficient depletion of TS following the insertion of a mutagenic lesion into the NTS, which is a superior way to generate a TLS substrate containing a gap in the DNA strand opposite to the lesion (Figure 1a and Appendix A).

The protein region encoded by nucleotide sequence confined by the existing Bpu10I nicking sites within the EGFP gene is important for efficient excited-state proton transfer in the immediate fluorophore environment and for correct structural organization of the folded protein [37,57,58,59]. Not surprisingly, we found that most (17 of 23) of the amino acid substitutions tested in the fluorescence screen had negative effects on the EFGP brightness; however, only six led to a complete loss of fluorescence (Figure 2 and not-shown data). Thus, a complete EGFP inactivation was only observed for mutations leading to a proline residue invariantly breaking the beta-sheet structure (T204P, Q205P, S206P, and A207P), a termination codon (Q205*) or a bulky amino acid residue on the fluorophore side of the beta-strand (S206Y). Consequently, we did not find any fully inactive EGFP mutant containing guanine at a nondegenerate codon site, which would have a potential to lead to an amino acid with changed properties upon a random secondary mutation. Thus, the S206A (c.616T>G) mutant retained approximately one third of the original EGFP brightness (Figure 3a) and was reversible to the wild-type phenotype by only one of the three possible base substitutions (data not shown). Nonetheless, with the exception of guanine, all other bases were covered by the identified set of mutants to meet the requirements for both the complete loss of EGFP fluorescence and the capacity of its restoration by any kind of subsequent base substitution (Figure 3). The newly identified mutants thereby provide a set of reporters for the detection of DNA TLS errors at DNA lesions derived from A, C or T. The same reporter vectors (but with a different procedure) are useful for analyses of TM induced by lesions affecting T, G or A, respectively.

To test the sensitivity of the new reporters, we generated constructs containing synthetic AP lesions at the positions deemed suitable for detection of erroneous DNA TLS and TM. Although it has long been known that the bypass of AP lesions frequently results in mutations [60], it is controversial whether AP sites are non-instructive or rather miscoding during DNA synthesis in mammalian cells. Thus, shuttle vectors recovered from the monkey COS-7 cells indicated a rather random pattern of nucleotide misincorporation opposite to the AP lesion, a slight bias against dG [38], or a preferential—but not strictly specific—incorporation of either dA or dT [39]. In contrast, studies conducted by different groups in human cells revealed clear preferences to dA followed by dT and dC [14,61]. Remarkably, a rapid quantitative estimate based on very different EGFP reversal rates documented in the Q205* and A207P mutants in our present study (compare Figure 1c and Figure 5) unmistakably indicated that incorporation of dA is strongly favored over dG in human cells. Moreover, since the fluorescence recovery rate in the A207P mutant approaches the maximum repair rate set by the conventional EGFP construct containing a single-stranded gap of the same size, we infer that dA is incorporated in >70% cases, which is coherent with results obtained by others in human cells using the shuttle vector sequencing techniques [14,61]. It is important to take into account that none of human Y-family DNA polymerases have a preference for dAMP when inserting a nucleotide opposite an AP site [9]. Considering that of the B-family polymerases, only pol α and pol δ can bypass AP site with relatively high efficiencies and selectivity to dAMP [9,62,63], the results imply that a great portion of the overall bypass capacity in human cells is attributable to pol δ and a much smaller part of it to the dedicated TLS polymerase(s).

As in the case of erroneous DNA TLS, TM was also detected with excellent sensitivity using the gain-of-function reporter principle. Because AP sites in cells are very efficiently processed by the BER pathway, which would prevent TM, we had to use a structurally analogous synthetic lesion with a non-cleavable linkage to prevent the excision. We previously showed that the erroneous bypass of such a BER-resistant AP lesion during transcription can be detected using the EGFP Q205* (c.613C>T) reporter with a signal strength at the level of 10–80% of the wild-type EGFP reporter, depending on the status of NER as the secondary repair pathway in human cells [30]. Here, we tested the Q205* and A207P mutants, which yield apurinic lesions (in contrast to physiologically less relevant apyrimidinic sites), in HeLa cells. Considering that the transcriptional bypass of AP lesions in human cells occurs predominantly with incorporation of A [64], the resulting base substitutions should be detectable in both Q205* and A207P reporters (Figure 3). Despite the NER proficiency of the host cell line, we detected a signal at the level of >30% wild-type EGFP in both reporter constructs (Figure 1b and Figure 4c), which indicates extremely high detection sensitivity of TM.

## 5. Conclusions

In summary, by using AP lesions as a model, we demonstrated that nucleotide substitutions, which arise from the erroneous bypass of DNA lesions during either DNA or RNA synthesis, can be efficiently detected in cells using the EGFP Q205* (c.613C>T) mutant as a reporter with a positive readout. The EGFP Q205* reporter is sensitive to TLS errors induced by modifications of thymine in the NTS and to TM caused by modifications of adenine in the TS. In order to create reporters tailored to detection of mutagenic TLS and TM induced by modifications affecting other bases in DNA, we further designed a set of appropriate loss-of-function mutations in the nucleotide sequence in the EGFP region suited for seamless incorporation of custom synthetic DNA modifications into either DNA strand. As a proof of performance of the generated reporters, we analyzed the fidelities of the bypass of mutagenic AP lesions incorporated at the mutation sites. The obtained results provided reliable quantitative estimates of the error rates induced by AP lesions during TLS and TM as well as quick clues about specificities of the arising nucleotide substitutions. Considering the broad use of EGFP as a readily detectable fluorescent probe, and taking into account the ease, robustness and versatility of the experimental procedures described here, we hope that our tools will be useful to the community for the characterization of miscoding properties of defined types of DNA modifications and the elucidation of the specific damage tolerance mechanisms of the cell.

## Figures and Tables

**Figure 1 biomolecules-10-00902-f001:**
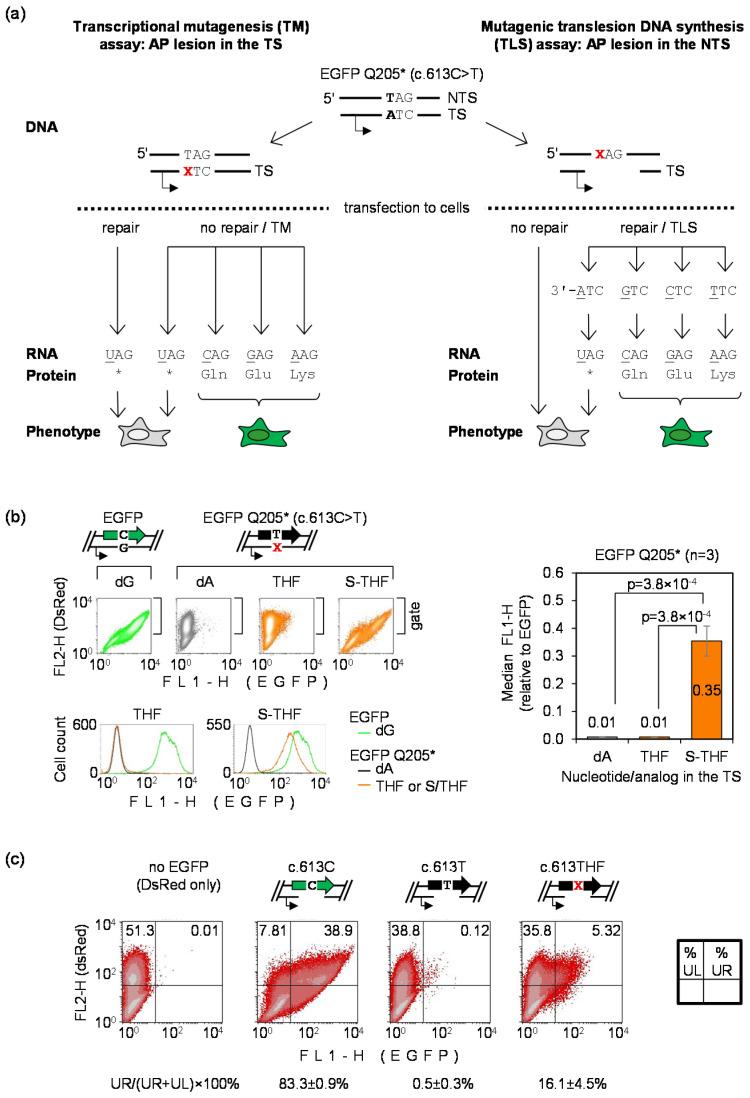
Transfection-based reporter assay for direct detection of transcriptional mutagenesis (TM) and mutagenic translesion DNA synthesis (TLS) templated by synthetic DNA lesions incorporated into vector DNA: (**a**) Scheme of the enhanced green fluorescent protein (EGFP) Q205* vector and of the derived constructs containing synthetic apurinic/apyrimidinic (AP) lesions (X) at the specified positions; (**b**) Detection of TM by flow cytometry of HeLa cells analyzed 24 h after transfection with constructs containing dA, THF and S-THF in double stranded DNA at the nucleotide 613 in TS. Scatter plots were gated by the DsRed fluorescence to generate the EGFP fluorescence distribution plots and calculate the median FL1-H. Grouped plots on the left show data of a representative experiment. Bar chart of the right shows quantification of the EGFP expression gain for *n* = 3 independent experiments (mean±SD); (**c**) Detection of mutagenic TLS. Cells transfected with constructs containing dT or THF at the nucleotide 613 in NTS opposite to a 18-nt gap were analyzed by flow cytometry 24 h post transfection. Mutant frequency was calculated as a ratio of EGFP-positive cells (in the upper right quadrant, UR) to total transfected cell count (UR+UL) for *n* = 3 independent experiments (mean ± SD).

**Figure 2 biomolecules-10-00902-f002:**
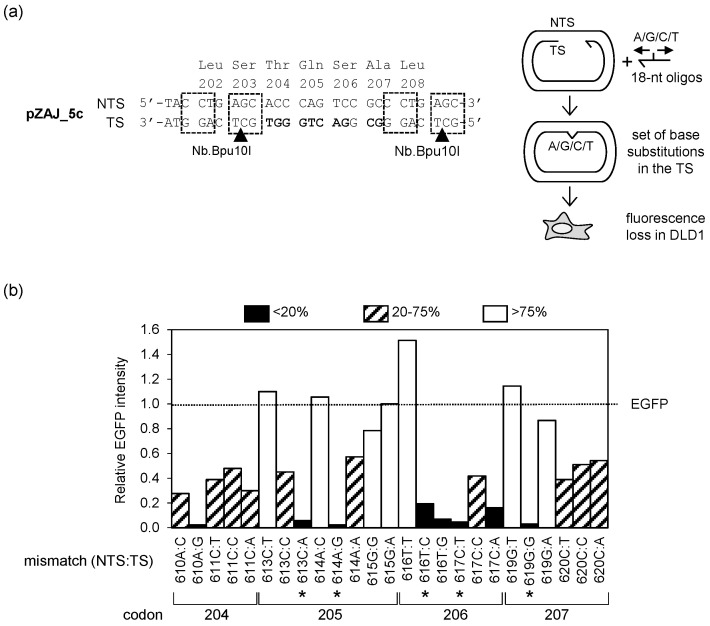
Phenotypic screening of mismatches with the indicated nucleotide substitutions in the TS to identify potential loss-of-fluorescence EGFP mutations: (**a**) Schematic representation of the screening strategy. The pZAJ_5c vector sequence fragment (on the left) shows tandem Nb.Bpu10I recognition sequences (boxes) and the nicking sites in the TS (arrowheads). Nucleotides targeted by single substitutions are marked bold; (**b**) Fluorescence intensities (relative to the original EGFP) measured in the DLD1 cells transfected with constructs containing the specified single-nucleotide mismatches (NTS:TS). Asterisks show candidate mutations selected for further validation.

**Figure 3 biomolecules-10-00902-f003:**
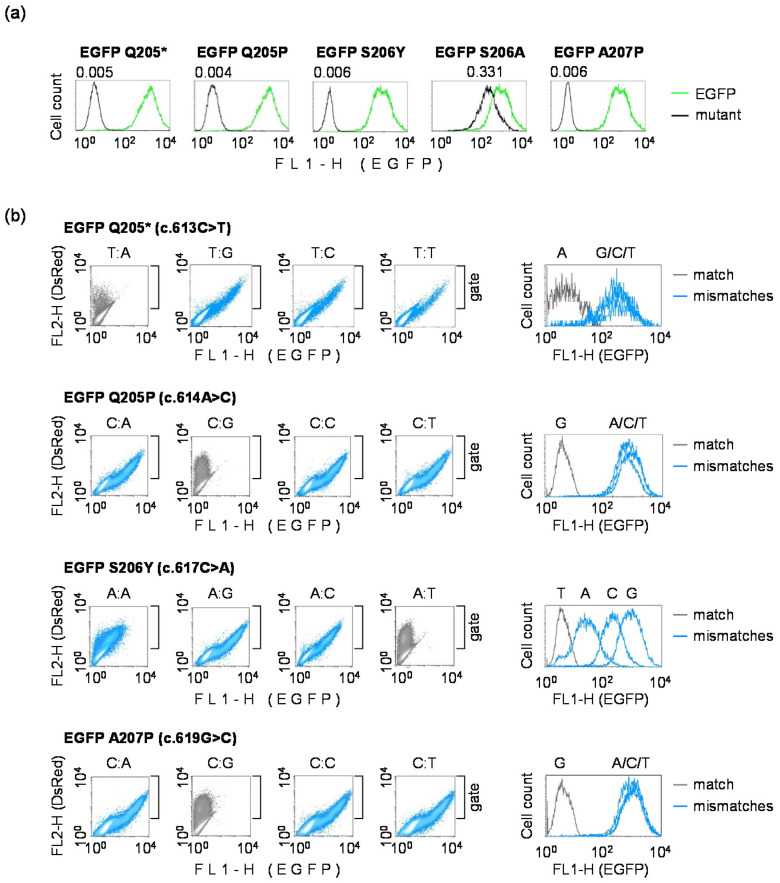
Validation of the candidate EGFP loss-of-function variants containing different nucleotide pairs at the mutation site: (**a**) Quantitative flow cytometry analyses of cloned EGFP mutants selected based on the low fluorescence signal in the preliminary screen. HeLa cells were transfected with vectors encoding the specified EGFP mutants or the original EGFP, in the presence of DsRed-Monomer as a transfection marker to generate the overlaid fluorescence distribution plots, as explained in Figure 1. Values above the peaks report fluorescence intensities relative to the original EGFP (calculated from the median FL1-H); (**b**) Reversal to the fluorescent phenotype by targeted single nucleotide substitutions in the TS. Expression vectors for the indicated EGFP mutants were used to generate all possible mismatches (blue color) or a correct base pair (grey color) by ligation of the respective synthetic oligonucleotides. EGFP fluorescence was analyzed in transfected DLD1 cells.

**Figure 4 biomolecules-10-00902-f004:**
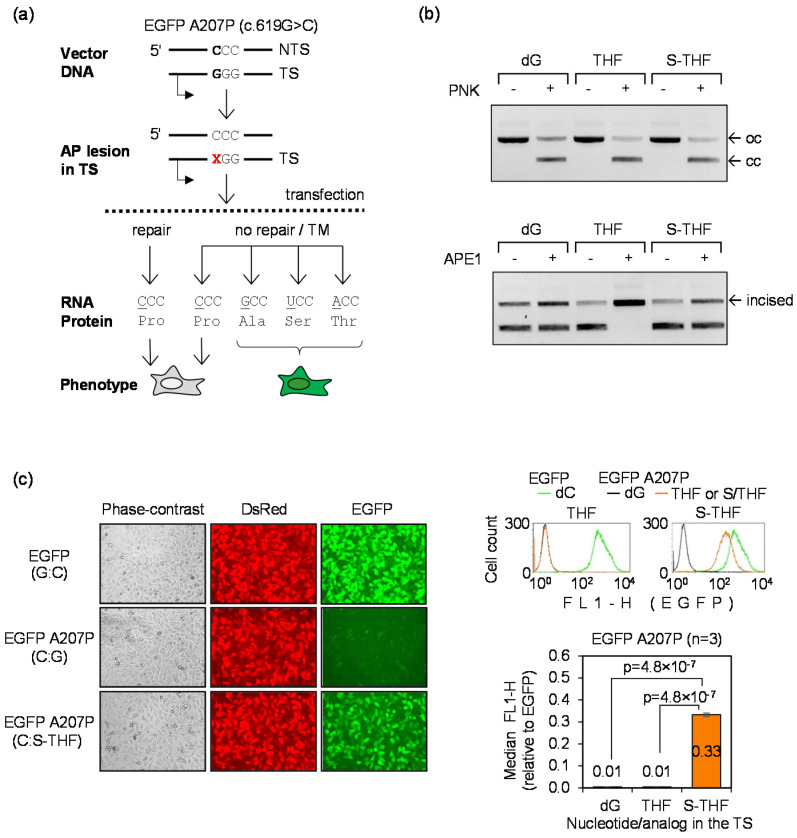
Detection of TM by AP lesion in the EGFP A207P (c.619G>C) reporter: (**a**) Scheme of the EGFP A207P reporter indicating position of the lesion (X) at the nucleotide 619 of the TS and possible outcomes of the transcriptional bypass; (**b**) Verification of incorporation of the lesions into the vector DNA. Upper gel: proof of incorporation of synthetic DNA strands containing dG or the indicated AP lesions (THF, S-THF) into the TS. The vector containing the single-stranded 18-nt gap in the TS migrates as open circular form (oc). Efficient ligation of synthetic oligonucleotides into the gap is confirmed by reconstitution of the covalently closed circular form (cc). Lower gel: verification of incision of the THF lesion by human APE1 and inhibited incision of the S-THF; (**c**) TM visualized by fluorescence microscopy and quantitative flow cytometry as a reversal of the non-fluorescent EGFP phenotype. HeLa cells were transfected with the indicated EGFP constructs in the presence of DsRed-Monomer as a transfection marker.

**Figure 5 biomolecules-10-00902-f005:**
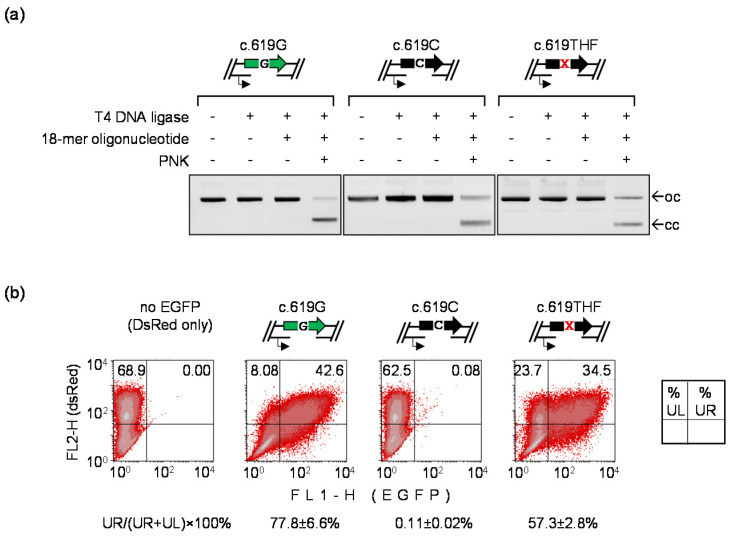
Detection of mutagenic TLS over the synthetic AP lesion in the EGFP A207P (c.619G>C) reporter by reversion to the fluorescent phenotype: (**a**) verification of the presence of gaps in the TS opposite to the specified nucleotides in the TLS reporter constructs by inhibited ligation and its reconstitution with matching synthetic 18-mer oligonucleotides. THF lesion at the nucleotide 619 is marked with “X”; (**b**) Repair of the 18-nt gap by re-synthesis of the missing TS fragment in the construct encoding the functional EGFP (c.619G) and detection of mutagenic TLS on the EGFP A207P template containing no modification (c.619C) or AP lesion (c.619THF) at the nucleotide 619. HeLa cells were transfected with the indicated constructs together with the DsRed-Monomer expression vector as a transfection marker. The fractions of EGFP-positive cells (UR/(UR+UL)×100%) were used as an estimate of the repair and mutation rates, as in Figure 1 (*n* = 2 independent experiments, mean ± range).

**Table 1 biomolecules-10-00902-t001:** Analysis of the EGFP coding sequence fragment flanked by the available Bpu10I sites and the results of preliminary screening based on the expression of mismatch-containing constructs in DLD1 cells. Candidates selected for validation and the Q205* mutation characterized previously are highlighted in bold typeface.

Codon No. and Amino Acid.	Codon (5′→3′)	Nucleotide Position and the Nature of the Mismatch (NTS:TS)	Expected Amino Acid Change	% EGFP Expression
204 Thr	ACC	610A:C	T204A	20–75%
		610A:G	T204P	<20%
		610A:A	T204S	no data ^1^
		611C:T	T204N	20–75%
		611C:C	T204S	20–75%
		611C:A	T204I	20–75%
		612C:T	=	not screened
		612C:C	=	not screened
		612C:A	=	not screened
205 Gln	CAG	613C:T	Q205K	>75%
		613C:C	Q205E	20–75%
		**613C:A**	**Q205** *	**<20%**
		614A:C	Q205R	>75%
		**614A:G**	**Q205P**	**<20%**
		614A:A	Q205L	20–75%
		615G:T	=	not screened
		615G:G	Q205H	>75%
		615G:A	Q205H	>75%
206 Ser	UCC	616T:T	S206T	>75% ^2^
		**616T:C**	**S206A**	**<20%**
		616T:G	S206P	<20%
		**617C:T**	**S206Y**	**<20%**
		617C:C	S206C	20–75%
		617C:A	S206F	<20%
		618C:T	=	not screened
		618C:C	=	not screened
		618C:A	=	not screened
207 Ala	GCC	619G:T	A207T	>75%
		**619G:G**	**A207P**	**<20%**
		619G:A	A207S	>75%
		620C:T	A207D	20–75%
		620C:C	A207G	20–75%
		620C:A	A207V	20–75%

^1^ The same T204S amino acid change is tolerated in the 611C:C construct; ^2^ Recorded fluorescence higher than in the original EFGP construct; (*) Termination codon; (=) No amino acid change expected.

**Table 2 biomolecules-10-00902-t002:** Summary of the loss-of-fluorescence EGFP mutants convertible to fluorescent protein species by the indicated single nucleotide substitutions (underlined) and EGFP phenotypes of the analyzed secondary mutations in the TS.

Non-Fluorescent Mutants			Detectable Secondary Mutations			
Protein	DNA	Base Pair (NTS:TS)	DNA TS 3′→5′ (TLS)	RNA 5′→3′ (TM)	Amino Acid Change ^1^	Brightness (% EGFP) ^2^
Q205 *	c.613C>T	T:A	GTC	CAG	*205Q	100
			CTC	GAG	*205E	~70
			TTC	AAG	*205K	~100
Q205P	c.614A>C	C:G	GTC	CAG	P205Q	100
			GAC	CUG	P205L	~80
			GCC	CGG	P205R	~100
S206Y	c.617C>A	A:T	AGG	UCC	Y206S	100
			AAG	UUC	Y206F	~3
			ACG	UGC	Y206C	~20
A207P	c.619G>C	C:G	CGG	GCC	P207A	100
			AGG	UCC	P207S	~100
			TGG	ACC	P207T	~100

^1^ Mutations restoring the original EGFP amino acid sequence are listed for each mutant in the first row; ^2^ Estimates based on the expression of constructs in which the TS fragment was replaced by synthetic DNA oligonucleotides containing the indicated mismatched nucleotides (underlined in the DNA sequence). For each mutant, signals of constructs with restored TS sequence of the wild-type EGFP were set as 100% and relative brightness of the remaining two mismatched constructs calculated accordingly. The estimates are based on results of parallel transfections performed in duplicates. The relative brightness values of constructs containing matched nucleotides (encoding mutant EGFP variants) were negligibly low in all cases (≤0.5%); (*) Termination codon.

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
