# Peer review of "EGFP Reporters for Direct and Sensitive Detection of Mutagenic Bypass of DNA Lesions"

_biomolecules, 2020, doi:10.3390/biom10060902_

Round 1
Reviewer 1 Report
The manuscript by Rodrigues-Alvarez et al. describes a system foe measuring mutation resulting fro translation synthesis (TLS) and from transcriptional mutagenesis (TM) a mechanism by which errors made during transcription allow expression of mutated proteins. They use a critical region of the GFP protein sequence that could have either the non-transcribed strand removed to measure TLS, or the transcribed strand remove to measure TM. They then ligated in an oligo containing the appropriate lesion for study. For TM, they constructed AP site that could not be corrected by BER so that onlt TM could restore green fluorescence. Initially, for validation, the authors studied a nonsense codon for which any substitution for T would restore the reading frame. They showed that, as expected, AP sites are usually filled by dA in HeLa cells. They also showed, amazingly, that TM is sufficiently frequent to give robust restoration of green fluorescence.
The authors proceed to mutate other bases in the critical region of the GFP sequence, and they found several substitutions that would prevent fluorescence that could be reversed by mutation. They obtained mutations that could be used to model A, T or C base lesion both during TLS and TM. Thus they have created a system that can be used to characterize the mutagenicity of a very broad spectrum of base lesions during either TLS or TM. This facility has not been available previously and has the potential to be a very useful tool for many investigations.
The study is very thorough and detailed, and this reviewer detected no problems in technology or in the results.
I would have preferred to have the logic of the TM assay explained clearly earlier, because how this was to be achieved was a problem from the beginning.
The language is generally good but please don't say "encoding for". "Encoding" means "coding for" (Line 162 and 201). On line 79, ulterior is surely the wrong word.
Author Response
Point 1: „I would have preferred to have the logic of the TM assay explained clearly earlier, because how this was to be achieved was a problem from the beginning.“
Response 1: We have re-written section 3.1. (on page 5), beginning with the explanation of the TM and TLS assays. Me have also modified the caption to Figure 1 accordingly.
Point 2: “The language is generally good but please don't say "encoding for". "Encoding" means "coding for" (Line 162 and 201). On line 79, ulterior is surely the wrong word.”
Response 2: We have corrected the wording, as suggested.
Reviewer 2 Report
The manuscript by Rodrigues-Alvarez describes the construction of several EGFP reporters for the measurement of translesion DNA synthesis (TLS). It is a natural extension of the work from this group published last year in Nucleic Acids Research (Kitsera et al reference#30). The work is carefully conducted and much of the data results from replicate independent experiments. The manuscript is well-written and they confirm the non-random preference for insertion of dA opposite an AP lesion.
The only real reservation I have with the work is that the EGFP reporters have not been tested in cells defective in the TLS polymerases (much as they used NER-defective cells in the Kitsera paper.) Such data would give robust validation of the system and strengthen the paper.
Of lesser importance is what seems to be the suggestion at the beginning of the discussion that the reporter assay may help trace mutations back to exposure to a particular carcionogen. I am not sure how this would work.
They may also with to compare their methodology with another recently published one (Choi JS., Berdis A. (2019) Artificial Nucleosides as Diagnostic Probes to Measure Translesion DNA Synthesis. In: Shank N. (eds) Non-Natural Nucleic Acids. Methods in Molecular Biology, vol 1973. Humana Press, New York, NY)
Author Response
Point 1: “The only real reservation I have with the work is that the EGFP reporters have not been tested in cells defective in the TLS polymerases (much as they used NER-defective cells in the Kitsera paper.) Such data would give robust validation of the system and strengthen the paper.”
Response 1: This is a valid point. We have currently started in-depth investigation of TLS of several DNA lesions (including various types of AP sites) using HAP1-derived knockout cell lines for Y-family DNA polymerases. We still need to solve several issues. One specific problem with AP lesions is that, contrary to initial expectations, most of bypass appears to originate from a replicative DNA polymerase (discussed on page 15, lines 536-545). Accordingly, we see only subtle differences between the TLS knockout cell lines.
Point 2: “Of lesser importance is what seems to be the suggestion at the beginning of the discussion that the reporter assay may help trace mutations back to exposure to a particular carcinogen. I am not sure how this would work.”
Response 2: We have re-written the fragment (lines 479-488 in the revised version) to explain this point more clearly.
Point 3: “They may also with to compare their methodology with another recently published one (Choi JS., Berdis A. (2019) Artificial Nucleosides as Diagnostic Probes to Measure Translesion DNA Synthesis. In: Shank N. (eds) Non-Natural Nucleic Acids. Methods in Molecular Biology, vol 1973. Humana Press, New York, NY)”
Response 3: We have considered the suggested paper, which describes synthetic NTP incorporated into DNA during replication as tools for quantitative assessment of TLS capacity of cells. However, in our manuscript we wanted to put more emphasis on the fidelity of bypass of modifications pre-existing in the template DNA and on the resulting mutagenesis. We found it difficult to smoothly integrate both topics into discussion of our results.
Reviewer 3 Report
This manuscript describes an improved plasmid-based tool for assaying translesion synthesis and transcriptional mutagenesis in human cells. Specifically, the authors have designed plasmids with an EGFP reporter gene that has been inactivated by single point mutations that inactivate EGFP and can revert back to a functional EGFP by multiple base substitutions. One of these plasmids (Q205STOP) was modified to include abasic sites in the transcribed or the non-transcribed strand and used to detect TM and TLS events, respectively, in vivo. The assay appears to be sufficiently sensitive and the validation appears thorough. However, there are some issues that would improve the manuscript:
- Since cancer cells upregulate TLS, the assay should also be performed in non-cancer cell lines. As currently performed, the study raises the question whether the high TLS rates are specific to the HeLa cancer cell line and what the usefulness and sensitivity of the assay would be in other cell lines, such as non-cancer cell lines or primary cells.
- Considering this relatively straight-forward assay, the manuscript, especially section 3.1., is unnecessarily cumbersome. Explaining more clearly early on in the text how TLS and TM deal with the two abasic site constructs would make this manscript (and the assay) more accessible to researchers not already TLS/TM experts.
- Related to the point above, it would be helpful if the authors explained in section 3.1 why the construct used for TM detection is not subject to TLS prior to being transcribed.
- Why other repair and tolerance pathways besides BER and TLS won't act on the abasic sites and gapped plasmid should be discussed. How does this affect the interpretation of the assay results.
- It's not clear how useful the presentation of the data in Figure 3b is. It might be easier to use a table, showing the quantification of the flow cytometry analysis in triplicate.
- It looks like there might be a mistake in the labeling of the S206Y mutant graph on the very right (shouldn't the left peak be the T not the G?)
- In Table 2, it is not clear from the footnote how the estimates in EGFP brightness (last column) were generated.
- The mutation of S206Y to S206F/C does not appear to restore fluorescence. This eliminates this mutation?
- It would strengthen the manuscript if there was a discussion of how relevant the BER-insensitive S-TFH lesion in the assay is to a natural (in vivo) substrate of TLS.
- A table that compares the capabilities of the available TLS/TM assays to the improved assay in this manuscript would be helpful.
Minor issues:
line 88: "a row of EGFP mutants" should be "a number of EGFP mutants"
line 201: delete "for"
Table 2: 4th column, 3rd row from the bottom: next to CGG there is an extra underscore
line 373/374: bracket issue, should be ([30] and unpublished results).
Throughout the manuscript there is an issue with missing articles, such as "the" and "a".
Author Response
Point 1: „Since cancer cells upregulate TLS, the assay should also be performed in non-cancer cell lines. As currently performed, the study raises the question whether the high TLS rates are specific to the HeLa cancer cell line and what the usefulness and sensitivity of the assay would be in other cell lines, such as non-cancer cell lines or primary cells.“
Response 1: This is a valid point. Indeed, mutation rates detected by our TLS reporter assay differ considerably between cell lines. Thus, in spite of good TLS proficiency, HeLa cells show lower than average rates of EGFP-positive mutants. We think, this is because other mechanisms, besides gap-filling DNA synthesis, significantly contribute to processing of the gapped vector DNA, resulting in mutations leading to non-fluorescent gene products. This is reflected by rather heterogeneous population of transfected cells in scatter plots. As requested, we now included new supplementary Figure S2 with representative data from non-cancerous MRC5 cell line. MRC5 cells show a more homogeneous distribution of EGFP fluorescence, which probably indicates priority of TLS over the alternative damage tolerance or repair mechanisms in this cell line. We have included description of the relevant results in the revised version (page 5, lines 268-274).
Point 2: „Considering this relatively straight-forward assay, the manuscript, especially section 3.1., is unnecessarily cumbersome. Explaining more clearly early on in the text how TLS and TM deal with the two abasic site constructs would make this manscript (and the assay) more accessible to researchers not already TLS/TM experts.“
Response 2: We have modified caption to Figure 1 and re-written section 3.1. (on page 5) to explain the assays more clearly.
Point 3: Related to the point above, it would be helpful if the authors explained in section 3.1 why the construct used for TM detection is not subject to TLS prior to being transcribed.
Response 3: We have clarified this very important point in the revised version (lines 210-215). To avoid TLS, it is critical to ensure that vector does not replicate in the host cell line. We controlled this but had forgotten to mention in the initial version.
Point 4: Why other repair and tolerance pathways besides BER and TLS won't act on the abasic sites and gapped plasmid should be discussed. How does this affect the interpretation of the assay results.
Response 4: We have now explained that action of repair and damage tolerance pathways (except TLS) would reduce the chance of specific base substitution mutations and conversion to fluorescent EGFP (lines 270-274).
Point 5: It's not clear how useful the presentation of the data in Figure 3b is. It might be easier to use a table, showing the quantification of the flow cytometry analysis in triplicate.
Response 5: In fact, values in the right column of Table 2 include quantification of data shown in Figure 3b. Nevertheless, we think that graphical representation helps to appreciate the magnitude of the registered positive signals. For instance, weakly fluorescent Y206F mutant is clearly distinguishable from the non-fluorescent protein on the fluorescence distribution plot.
Point 6: It looks like there might be a mistake in the labeling of the S206Y mutant graph on the very right (shouldn't the left peak be the T not the G?)
Response 6: Yes! We have corrected the label.
Point 7: In Table 2, it is not clear from the footnote how the estimates in EGFP brightness (last column) were generated.
Response 7: We have explained the calculation in the revised version (lines 389-394).
Point 8: The mutation of S206Y to S206F/C does not appear to restore fluorescence. This eliminates this mutation?
Response 8: We think that both S206F and S206C mutants are still useful. With >20% residual fluorescence, S206C certainly is. Even the very weakly fluorescent (at 3% EGFP) S206F mutant is clearly detectable above the background (as seen in Figure 3b). Of course, the sensitivity of detection of these mutations would be proportionally lower. We have added a more detailed description of outcomes of the individual mutations and discussed the limitations of the S206Y mutant (lines 368-369 and 371-380).
Point 9: It would strengthen the manuscript if there was a discussion of how relevant the BER-insensitive S-TFH lesion in the assay is to a natural (in vivo) substrate of TLS.
Response 9: We agree, considering that THF and S-THF are chemically different from naturally occurring AP sites. On the other hand, they are very similar structurally. Besides, we would like to mention that our other TLS data (unpublished) for the “natural” AP lesion (obtained by excision of uracil) reproduces very closely the effects of THF. We felt that focusing at this point would extend discussion beyond the focus of the paper.
Point 10: A table that compares the capabilities of the available TLS/TM assays to the improved assay in this manuscript would be helpful.
Response 10: We agree in principle. However, a comprehensive compilation of advantages and drawbacks of all published assays into a compact and uniformly formatted table was hardly possible and also hardly compatible with the experimental paper format. After careful re-assessment of the related work referenced in the manuscript, the authors agreed that the most important features of the assays described previously are adequately highlighted in the manuscript (lines 53-61; 65-69; 490-495; 530-534; 554-555). Also, in our opinion, the advantages of our system in comparison to others are sufficiently discussed (lines 495-505; 560-567).
Minor issues
Response: We undertook all suggested corrections
Reviewer 4 Report
In this work, Rodriguez-Alvarez et al aimed at generating a series of reporters to measure directly the miscoding capacities during Translesion DNA synthesis (TLS) or RNA synthesis in the presence of a damaged DNA in human cells. As reporter gene they used the enhanced green fluorescent protein (EGFP), and as a proof of concept they used a previously identified mutant EGFPQ205* that gives a non-fluorescent truncated protein. Since any ulterior base substitution at the primary mutation (at the position 613) will restore a functional fluorescent EGFP, they can directly score for TLS and transcriptional mutagenesis (TM).
They cleverly exploit the fact that the nucleotide sequence of the EGFP containing amino-acid 205 and other amino-acids involved in the fluorescence active site of EGFP, are flanked by the Bpu10I sites. This strand-specific nicking endonuclease allow introducing DNA modifications either in the transcribed or non-transcribed strand. Therefore, it is possible to use the reporter either to measure TLS (when the lesion is positioned in the non-transcribed strand) or TM (when the lesion is positioned in the transcribed strand).
Following a phenotypic screening, they identified other loss-of function EGFP mutants that can be used as reporters to measure directly TLS and TM when different types of base pairs are involved.
Having an easy and direct read-out of mutagenesis during replication or transcription of a damaged DNA is very important to better investigate genome instability mechanisms. Therefore, the set of tools described in this study can be of high interest for the scientific community working on genome instability and DNA damage field. Overall, this work is very interesting, the manuscript well written and the data are of good quality. However, I do not completely agree with some of the statements and have some suggestions that might help improving the quality of the paper:
- the authors tested their reporters only under physiological conditions and used only one DNA modification (an abasic site). I would have liked the authors to challenge a bit more their reporter systems by using more than one DNA lesion and/or testing some genotoxic stress conditions. This would strengthen their study and prove the robustness of their reporter systems.
- in the same line, it would have been nice to see the method applied to a scientific question, like for instance by knocking down a TLS polymerase and assess the effect on TLS.
- I do not agree with the statement lines 346-347. The result for the mutant EGFP S206Y (Fig3b) shows that only the substitution of A:G fully restores EGFP signal. And this is also shown in the Table 2 with the brightness %. Therefore, I do not fully agree with the statement of lines 352-353: the set of reporters is not fully complete. The authors should explain this or modulate their statement.
- I am not very convinced of the author's conclusion on line 400. They got only 16% (and not a marked regain) of the population with a functional EGFP when they measure TLS using the EGFP Q205* mutant, which seems to confirm the “A rule”. Indeed, if the repair of the gap in which there is an abasic site is obtained inserting an A, the EGFP obtained will be truncated (i.e. no fluorescence).
The result obtained measuring TLS in the EGFP A207P mutant strengthens the idea that in front of an abasic side dA are favored.
- I was wondering why the authors did not sequence the vectors after TM or TLS event. This is a simple experiment to implement that would allow to have a better idea of the spectrum of mutations at the abasic site, especially for the TM events.
- from the manuscript I could not understand whether the vector pZAJ in which the EGFP gene is cloned is a replicative vector. This could affect the results for the TM measurements. This should be clarified.
Minor points:
Line 68: “TM would cause” instead of “TM caused”
Line 349: “and 619C:G in S207P”
In Fig 3b on the right graph for EGFP S206Y there is a mistake in the letters. There are two G instead of a T for the gray curve.
Fig 4b and Fig 5a could be moved to the Supplementary Figures since they are just control of the constructs and do not bring any important information to the paper.
Line 400: (Figure 1c)
Author Response
Point 1: „the authors tested their reporters only under physiological conditions and used only one DNA modification (an abasic site). I would have liked the authors to challenge a bit more their reporter systems by using more than one DNA lesion and/or testing some genotoxic stress conditions. This would strengthen their study and prove the robustness of their reporter systems.“
Response 1: We agree and we are actively working with several modifications. In particular, the Q205P and A207P perform spectacularly well in detection of TM by 8-oxoguanine and O6-methylguanine. Unfortunately, however, the TLS reporters currently available are not suited for guanine modifications. It is for this reason that we have chosen AP site as a lesion suitable for detection of both TM and TLS errors.
Point 2: „in the same line, it would have been nice to see the method applied to a scientific question, like for instance by knocking down a TLS polymerase and assess the effect on TLS.“
Response 2: We agree, of course. That was the original idea with AP lesions. However, we have realised that most of the bypass is done by a replicative DNA polymerase (discussed on page 15, lines 536-545). We already have data from the Y-family polymerases knockout cell lines, where we see only subtle differences from the wild-type. We are undertaking deep sequencing to clarify the contributions of individual polymerases. Unfortunately, current results are too preliminary to be accommodated in the present manuscript.
Point 3: „I do not agree with the statement lines 346-347. The result for the mutant EGFP S206Y (Fig3b) shows that only the substitution of A:G fully restores EGFP signal. And this is also shown in the Table 2 with the brightness %. Therefore, I do not fully agree with the statement of lines 352-353: the set of reporters is not fully complete. The authors should explain this or modulate their statement.“
Response 3: We have re-written the fragment (lines 371-380) and discussed the limitations of the S206Y mutant.
Point 4: „I am not very convinced of the author's conclusion on line 400. They got only 16% (and not a marked regain) of the population with a functional EGFP when they measure TLS using the EGFP Q205* mutant, which seems to confirm the “A rule”. Indeed, if the repair of the gap in which there is an abasic site is obtained inserting an A, the EGFP obtained will be truncated (i.e. no fluorescence).
The result obtained measuring TLS in the EGFP A207P mutant strengthens the idea that in front of an abasic side dA are favored.“
Response 4: Indeed, the results indicate that A is favored. However, we also have to point out that A is not the only nucleotide incorporated. We propose that a replicative polymerase incorporates A and TLS polymerases incorporate other nucleotides. We have modified the text to explain this (lines 537-545).
We would like to add that, based on our preliminary sequencing results in several cell lines, we estimate 70 to 85% of adenine. Sequencing data from TLS knockout cells should clarify which polymerase makes the rest.
Point 5: „I was wondering why the authors did not sequence the vectors after TM or TLS event. This is a simple experiment to implement that would allow to have a better idea of the spectrum of mutations at the abasic site, especially for the TM events.“
Response 5: There are some potential culprits with sequencing vector DNA recovered from cells. For instance, we realised that it is very hard to rule out that mutation is induced by DNA polymerase used for the library preparation (when DNA lesion is not repaired in cells). We think, this is the major source of discrepancy in published reports. We are still optimising our sequencing protocols towards possibly clean results.
Point 6: „from the manuscript I could not understand whether the vector pZAJ in which the EGFP gene is cloned is a replicative vector. This could affect the results for the TM measurements. This should be clarified.“
Response 6: This is extremely important point, which we had forgotten to mention in the initially submitted version. For the assay to work, it is essential that vector does not replicate in the host cell line. We have clarified this in the revised version (lines 210-215).
Minor points
Response: We have corrected the erroneous label in Figure 3b and done other minor corrections.
Round 2
Reviewer 2 Report
The authors have addressed my comments satisfactorily
Reviewer 3 Report
The authors have addressed all issues satisfactorily.